# Structure-Based and Rational Design of a Hepatitis C Virus Vaccine

**DOI:** 10.3390/v13050837

**Published:** 2021-05-05

**Authors:** Johnathan D. Guest, Brian G. Pierce

**Affiliations:** 1University of Maryland Institute for Bioscience and Biotechnology Research, Rockville, MD 20850, USA; jdguest@terpmail.umd.edu; 2Department of Cell Biology and Molecular Genetics, University of Maryland, College Park, MD 20742, USA

**Keywords:** HCV, E1E2, structure-based vaccine design

## Abstract

A hepatitis C virus (HCV) vaccine is a critical yet unfulfilled step in addressing the global disease burden of HCV. While decades of research have led to numerous clinical and pre-clinical vaccine candidates, these efforts have been hindered by factors including HCV antigenic variability and immune evasion. Structure-based and rational vaccine design approaches have capitalized on insights regarding the immune response to HCV and the structures of antibody-bound envelope glycoproteins. Despite successes with other viruses, designing an immunogen based on HCV glycoproteins that can elicit broadly protective immunity against HCV infection is an ongoing challenge. Here, we describe HCV vaccine design approaches where immunogens were selected and optimized through analysis of available structures, identification of conserved epitopes targeted by neutralizing antibodies, or both. Several designs have elicited immune responses against HCV in vivo, revealing correlates of HCV antigen immunogenicity and breadth of induced responses. Recent studies have elucidated the functional, dynamic and immunological features of key regions of the viral envelope glycoproteins, which can inform next-generation immunogen design efforts. These insights and design strategies represent promising pathways to HCV vaccine development, which can be further informed by successful immunogen designs generated for other viruses.

## 1. Introduction

Hepatitis C virus (HCV) represents a global disease burden, with approximately 71 million people infected [1]. The majority of untreated HCV infections become chronic [2,3] and may lead to cirrhosis or hepatocellular carcinoma (HCC), a deadly type of liver cancer [4,5]. Although direct-acting antiviral (DAA) drugs have cure rates greater than 90% [6,7], they do not prevent a recurrence of HCV infection [8] and may not reduce the risk of HCC [9,10]. Combined with financial barriers and the asymptomatic nature of many HCV infections [11,12], treatment with DAAs alone has not been enough to stop HCV transmission, and development of an effective vaccine for HCV is still viewed as essential [13,14]. However, efforts to produce an HCV vaccine, many of which have been described in previous reviews [15,16,17,18,19], have thus far been unsuccessful. Multiple factors likely contribute to the difficulty in developing an HCV vaccine [20,21], including substantial diversity between genotypes [22,23], viral mutation in infected individuals leading to quasispecies that can escape neutralizing antibodies [24], epitope shielding by glycans on the E1 and E2 envelope proteins [25,26], epitope shielding by apolipoproteins in HCV lipo-viral-particles (LVPs) [27,28,29], and other mechanisms of immune evasion [30,31]. Current limitations of and lack of standardization for in vitro and in vivo models of HCV infection may also hinder the evaluation and comparison of vaccine candidates [13,32]. Additionally, a high-resolution structure of the E1E2 glycoprotein complex, which is the target of neutralizing antibodies against HCV and thought to be a trimer of heterodimers on the surface of the virion [33], has not yet been determined, due in part to structural flexibility [34] and the requirement of hydrophobic transmembrane domains for assembly [35,36]. Structural characterization of envelope glycoprotein assemblies for other viruses has been facilitated by a trimerization domain as a scaffold [37,38], a modified furin cleavage site [39], or targeted stabilizing mutations [40,41,42], enabling structure-based vaccine designs for those antigens [43,44]. Remarkable progress was achieved even in human immunodeficiency virus (HIV) despite challenges of diversity, flexibility, and glycan shielding in the Env glycoproteins [45,46,47] that are broadly similar to challenges observed for HCV and E1E2.

Though the structure of the E1E2 heterodimer is not known, broadly neutralizing antibody (bnAb) interactions with E1 and E2 have been structurally characterized, providing insights into the neutralization determinants of known epitopes that may be crucial for stimulating protective B cell responses [48,49]. Conserved clusters of epitopes on E2 have been classified either as antigenic domains A-E (nomenclature used for this review) [50,51,52], epitopes I–III [20], or antigenic regions (ARs) 1–3 [53], and the AR classification also includes E1E2 epitopes (AR4, AR5) [54]. Although different epitope clusters can overlap [31,55], epitope mapping and structural studies have identified the following key E2 regions for bnAb recognition: antigenic domain B (residues 529–535 in H77 isolate numbering), domain D (residues 434–446), and domain E (residues 412–423), all of which contain residues critical for antibody binding that are nearly or fully conserved across genotypes [56,57]. Antibodies targeting these three antigenic domains of E2 neutralize the virus through competition with CD81, an HCV co-receptor that is critical for viral entry [58,59,60]. Conserved epitopes targeted by bnAbs have also been mapped to E1 (residues 314–324) [61] and the E1E2 heterodimer; the latter epitopes collectively include hotspot residues in both the stem region of E2 and the N-terminus of E1 [49,54,62]. On the other hand, antigenic domain A, which has been mapped to a region that includes E2 (residues 627–637) [31], is considered unnecessary for eliciting bnAbs, as antibodies that target this epitope are non-neutralizing [63].

Structure-based HCV vaccine design can be guided by this information, as sequence and structural determinants for many bnAb-associated epitopes are well known [55,56,64], with the notable exceptions of bnAb epitopes on the E1E2 heterodimer and the structural context of E1 and E2 epitopes that would be provided by the E1E2 ectodomain structure. However, large gaps in the structural knowledge of E1E2 have led to designs of the heterodimer that increase stability and solubility while maintaining or improving immunogenicity. In this review, we summarize recent structure-based and rational vaccine design efforts that harness current structural and antigenic knowledge of HCV glycoproteins to generate improved immunogens and, in some cases, to help elucidate the ectodomain structure of E1E2. The path to an effective HCV vaccine may include utilization of design strategies that generated promising immunogens for other viruses, including HIV, RSV, and SARS-CoV-2 [38,39,40,41,42], which have led to clinical-stage candidates [65,66,67] and successful vaccines [68,69].

## 2. Design Approaches

### 2.1. Epitope Scaffolding and Epitope-Based Designs

Numerous structure-based HCV vaccine design efforts have focused on utilizing conserved glycoprotein epitopes as immunogens and stabilizing one or more of these epitopes, either through scaffolding or cyclization. These strategies attempt to focus responses to key epitopes and have been employed for RSV and HIV immunogen designs to elicit more neutralizing antibodies than the unmodified proteins [38,70]. This design approach, along with others described in this review, is shown in Figure 1. Targets of epitope-based designs have included domain E (also referred to as Epitope I or AS412), a highly conserved linear epitope near the N-terminus of E2 that has been structurally characterized in complex with multiple bnAbs [71,72,73,74]. In many cases, scaffolded and epitope-based designs have been tested for immunogenicity and elicitation of neutralizing antibodies in vivo; these studies are described along with other in vivo-tested HCV immunogen designs in Table 1. 

Informed by the β-hairpin conformation of the domain E epitope found in multiple bnAb-bound structures [71,72], Sandomenico et al. designed a cyclized version named C-Epitope I, with E2 residues 411 and 423 mutated to cysteines that form a disulfide bridge to present epitope residues 412–422 [75]. This cyclized peptide was injected into mice following conjugation to a carrier protein, which was either keyhole limpet hemocyanin (KLH) or bovine serum albumin (BSA). Seven monoclonal antibodies that recognized C-Epitope I were isolated from murine hybridomas and showed pronounced binding specificity for the cyclic peptide design. Some of these isolated antibodies bound to the native epitope in the context of E2 with nanomolar affinity, but were found to be non-neutralizing. The structure of C-Epitope I was determined in complex with one of the induced murine monoclonal antibodies, then compared to the linear peptide bound to bnAb AP33 [71]. This comparison suggested that the conformationally-restrained C-Epitope I may not recapitulate the native β-hairpin, in part because a domain E residue that is typically glycosylated (N417) [25] was partially buried in the cyclized peptide interface with the monoclonal antibody [75]. 

A similar study generated designs of cyclized domain E peptides, but instead utilized the Protein Data Bank (PDB) [76] to compare the β-hairpin conformation with other protein structures in order to find a template for a cyclized peptide that would mimic the bnAb-bound epitope [77]. These designed immunogens bound to the neutralizing antibody HCV1 [72], and a crystal structure of HCV1 in complex with one cyclized peptide, named C1, confirmed that the native β-hairpin conformation and key bnAb interactions were retained as designed. Serum neutralization levels induced by these designs were relatively limited, but the cyclic peptides were still found to be immunogenic and induced antibodies that neutralized H77 HCV pseudoparticles (HCVpp), representing an improvement in immunogenicity over the linear (non-cyclic) peptide control [77]. Aside from cyclization modes that were different from the C-Epitope I design reported by Sandomenico et al., it is worth noting that the domain E peptides used in the Pierce et al. study included an N-glycan at position N417 (reflective of the glycan in native E2), which may have helped to induce the detectable, but low, neutralizing antibody responses in mice. Another study searched for structures that could provide a scaffold for either the domain E epitope or an E1 epitope, which adopts a helical conformation when bound to bnAb IGH526 [61,78]. For each epitope, the bnAb-bound conformations were compared to a large database of structures from the PDB, with structural similarities evaluated by six alignment algorithms [78]. Following this selection process, each target epitope was grafted onto a structurally similar scaffold and optimized, effectively presenting these HCV epitopes in a new context. Designs of scaffolded domain E and IGH526 epitopes were recognized by bnAbs known to target the native epitopes, and their antigenicity was increased through multimeric display of select scaffolded designs with self-assembling ferritin nanoparticles [78].
Figure 1Structure-based approaches for HCV E1E2 vaccine design. Approaches shown represent: (1) scaffolding of epitopes from E1 and E2, (2) design of the E2 antigen through truncation or residue substitutions to alter antigenicity, immunogenicity, or epitope exposure, and (3) scaffolding of E1E2 to generate stable, secreted glycoproteins. These design strategies are labeled as “Epitope scaffolding”, “Antigen design”, and “Antigen scaffolds”, respectively. Molecular structures shown are from PDB codes 4N0Y (E1 epitope) [61], 5KZP (E2 epitope) [77], 4UOI (E1 N-terminal ectodomain) [79], 6MEI (E2 ectodomain core) [80], 1FOS (Fos-Jun E1E2 scaffold) [81] and 6DMA (DHD15 E1E2 scaffold) [82], and were rendered in PyMOL v. 1.8 (Schrödinger, LLC). Red points on E2 represent rationally selected modifications of E2 (residue substitutions or loop truncations). Residue ranges for ectodomains and transmembrane domains are labeled according to H77 numbering.
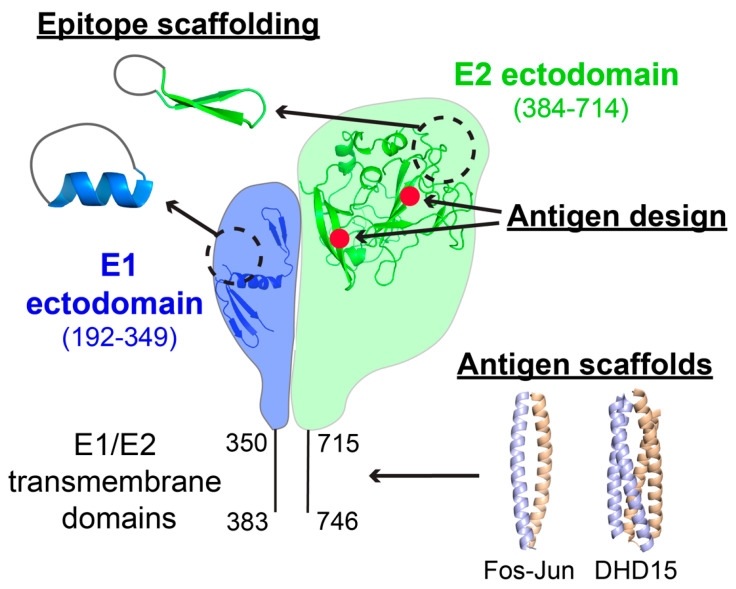


Several other studies have reported and characterized epitope-based HCV vaccine designs that do not utilize structure-based design or scaffolding of the epitopes themselves. Recently, Cowton et al. generated and structurally characterized an anti-idiotype monoclonal antibody, named B2.1A, that was recognized by AP33, effectively designing a structural mimic of domain E in a β-hairpin conformation [83]. Antibodies induced by B2.1A protected against HCV challenge from genotype 1 or 2 strains in an immunosuppressed and humanized liver chimeric mouse model, suggesting that B2.1A could form the basis of distinct immunogen designs focused on antigenic domain E [83]. The development of synthetic peptide libraries derived from HCV E1E2 found several peptides similar to hypervariable region 1 (HVR1) [84], domain E, and domain D that exhibited robust antigenicity in ELISA and elicited neutralizing murine antibodies in vivo when linked to a T-helper cell epitope [85,86]. Another peptide vaccine that combined the epitope from IGH526-bound E1, epitopes from E2 domains C and D, and peptides from nonstructural HCV proteins showed stimulation of humoral and cellular immunity in mice [87]. Cyclic peptides that are recognized by domain D antibodies have been generated through immobilization by synthetic conjugation, effectively mimicking a discontinuous and conformational epitope [88]. As reviewed by others [89], multiple studies have capitalized on self-assembly of hepatitis B virus (HBV) capsid proteins [90] to insert epitopes from domain E [91,92], HVR1 [93,94,95], and additional E2 antigenic domains [96] into exposed loops of the HBV capsid protein. These designs formed chimeric and stable virus-like particles (VLPs), and a few designs elicited immune responses with some capacity to neutralize multiple genotypes [91,96].
viruses-13-00837-t001_Table 1Table 1E2 and E1E2 immunogen designs tested in vivo.Design(s)In Vivo DataViruses Tested for Neutralization ^1^Results ^2^ReferenceCyclized domain E peptides, truncated E2 core with additional domain E and HVR1 removedMouseHomologous HCVpp: H77Heterologous HCVpp: genotypes 1b, 4aNeutralized H77 in HCVpp assays[77]E2 core with proline mutations in domain D or E, glycosylation of domain A, HVR1 removedMouseHomologous HCVpp: H77Heterologous HCVpp: genotypes 1b, 2a, 2i, 4aIncreased neutralization of heterologous strains in HCVpp assays[97]E2 core with truncated and designed HVR2, displayed on nanoparticlesMouseHomologous HCVpp: H77Heterologous HCVpp: genotypes 1a, 2a, 5aIncreased neutralization of H77 in HCVpp assays[98]Cyclized and conjugated domain E peptideMouseHomologous HCVcc: H77Heterologous viruses: not testedAntibody isolated from mice non-neutralizing in HCVcc assays[75]E2 core with eight cysteines mutatedMouseHomologous HCVpp: H77Heterologous viruses: not testedNo neutralization of H77 in HCVpp assays[99]E1E2 ectodomains oligomerized with C4b-binding protein IMX313PMouseHomologous virus: not testedHeterologous HCVpp: genotypes 1a, 1b, 2a, 2b, 3a, 4, 5, 6Neutralized most strains in HCVpp assays[100]E2 core with HVR1 and HVR2 removed, IgVR deglycosylated, and displayed on nanoparticlesMouseHomologous HCVpp: H77Heterologous HCVpp: genotype 1bIncreased neutralization of H77 in HCVpp assays[101]E2 core fused to ferritinMouseHomologous HCVcc: Con1Heterologous HCVcc: genotypes 1a, 1b, 2a, 3a, 4a, 5a, 7aIncreased neutralization of strains in HCVcc assays[102]Synthetic consensus of E2 core from genotype 1 sequencesGuinea pigHomologous HCVpp: NotC1Heterologous HCVpp: genotypes 1a, 1b, 2a, 3Increased neutralization of H77 in HCVpp assays[103]E2 core with variable regions removed and seven cysteines mutatedGuinea pigHomologous HCVpp: H77Heterologous HCVcc: genotypes 2a, 3a, 5aNeutralized H77 HCVpp, limited neutralization of HCVcc[104]Combination of two HVR1 peptidesMouseHomologous HCVpp: C47Heterologous HCVpp: genotypes 1a, 1b, 2a, 3a, 4a, 5a, 6aNeutralized strains in HCVpp assays[105]E2 with mutated N-glycan sites and HVR1 removedMouseHomologous HCVcc: Jc1Heterologous HCVcc: genotype 5aHeterologous HCVpp: genotype 1aNeutralized Jc1 HCVcc, H77 HCVpp[106]E2 epitopes from domains B, D, and E, and HVR1 mimotope displayed on HBV-S VLPsMouseHomologous virus: not testedHeterologous HCVpp and HCVcc: genotypes 1a, 1b, 2aNeutralized strains in HCVpp and HCVcc assays[91]E2 epitopes from domains B-E displayed on HBV-S VLPsMouseHomologous virus: not testedHeterologous HCVcc: genotypes 1a, 1b, 2a, 2b, 3a, 4a, 5a, 6aIgGs purified from sera neutralized some strains in HCVcc assays[96]E1E2 ectodomains with C-terminal leucine zipper as scaffold, furin cleavage siteMouseHomologous HCVpp: H77Heterologous viruses: not testedNeutralized H77 in HCVpp assays[107]^1^ Homologous virus is listed by strain name; heterologous viruses are listed by genotype and subtype when known. ^2^ Summary of measured neutralization induced by design, or improvement in neutralization over non-designed control, if measured by the authors. Unless otherwise noted, results refer to serum neutralization.

### 2.2. E2 Antigen Design

Though the E2 ectodomain core (residues 384–661; also defined as residues 384–645 [99]) can be expressed as an immunogen alone [77,108,109], its wild-type form may not be optimal for generating bnAb responses, due to possible factors noted earlier such as occlusion of bnAb epitopes with glycans, glycoprotein dynamics, and the presence of non-neutralizing epitopes. Insights regarding the structural basis of bnAb recognition provide an opportunity to modify specific epitopes in the context of E2 to improve immunogenicity. The aforementioned Pierce et al. study identified structural similarity between the β-hairpin conformation of domain E and a region of domain A, which as noted above is associated with non-neutralizing antibodies, and replaced the domain A residues with a copy of the domain E epitope in two E2 core designs, but found no significant differences in neutralization potency compared to an unmodified E2 core when tested for immunogenicity in mice [77]. Another study engineered E2 to improve immunogenicity by modifying domain A and D epitopes [97]. First, a proline mutation was introduced at residue 445 in domain D to stabilize the epitope in a conformation recognized by domain D antibody HC84.26.WH.5DL [97,110]. After another analysis of E2 core structure, an N-glycosylation site was added at residue 632, masking the domain A site to decrease the elicitation of non-neutralizing antibodies targeting that region [97]. Though the proline substitution increased HC84.26.WH.5DL binding affinity and the N-glycan addition greatly reduced domain A antibody binding, only the proline substitution led to both increased neutralization of multiple HCVpp representing neutralization-resistant HCV strains and maintained neutralization of homologous and other heterologous HCVpp [97]. Additional research has focused on optimizing the E2 core to increase antigenicity and stability. Optimized E2 cores were designed using sequences derived from genotype 1 or genotype 6 strains [98] and were structurally characterized in complex with multiple bnAbs [80,111,112]. These optimized cores improved on previous constructs [80,111] by truncating and designing flexible HVR2 and β-sandwich loops, preserving E2 core antigenicity while increasing thermostability [98]. Top designs of E2 core were also displayed on ferritin and other nanoparticle assemblies, exhibiting improved immunogenicity [98]. Other studies have tested multivalent presentation of E2 core on nanoparticles. Display of E2 core on lipid-based vesicles [101] or ferritin [102] has led to higher neutralization of HCVpp or HCV cell-cultured virus (HCVcc) [113] and stronger binding to neutralizing antibodies than E2 core alone, demonstrating the benefit of multimeric display for optimizing immunogenicity. E2 core aggregation through disulfide cross-linking was addressed in another design, where removal of eight cysteine residues resulted in the production of monomeric E2 core [99]. However, the monomeric E2 core with cysteines removed did not stimulate T cell responses as well as the wild type, aggregated E2 core, and no groups of mice immunized with the design or other constructs showed detectable viral neutralization [99].

Other efforts to improve E2 antigenicity and immunogenicity have optimized sequences and removed flexible variable regions, offering a complement to structure-based designs. Based on sequence analysis, primarily of genotype 1a, synthetic consensus sequences of E2 have been reported, but have not consistently improved immunogenicity [103,114]. HVR1 has been removed in many E2 core designs [77,101,104,115], as it can interfere with and enable evasion of bnAb responses [116,117]. In one case, HVR1 was found to determine sensitivity to heterologous neutralization by sera from goats immunized with an E1E2 vaccine candidate, in part through differential binding with co-receptor SR-BI [118]. Along with other variable regions, HVR1 was removed in an E2 core design that maintained binding to CD81 and formed an apparent homodimer [119,120]. A high-molecular-weight form of this E2 core design showed promising immunogenicity in guinea pigs [121], and has been further optimized through mutation of cysteine residues [104]. However, removing HVR1 alone does not appear to improve the immunogenicity of vaccine designs [115,122]. Removal of HVR1 also had minimal effects on antigenicity and immunogenicity when combined with mutations on E2 core that either removed glycans or modulated recognition of antigenic domains A and D [97,106]. Counterintuitively, a bivalent peptide vaccine designed from HVR1 has induced neutralization of heterologous HCV strains [105].

### 2.3. E1E2 Optimization

While some design efforts have focused on presenting specific neutralizing epitopes, other groups have pursued larger and more complex scaffolds to optimize native-like assembly of E1E2 proteins, often by removing or replacing hydrophobic transmembrane domains [123] to produce a stable and soluble vaccine candidate. Cao et al. designed and characterized E1E2 constructs with either a C-terminal Fc-tag or a designed heterodimeric scaffold to enforce heterodimerization of soluble E1E2 and enable structural and functional characterization of E1E2, which has been hindered by the difficulties in producing that complex in homogenous form [124]. These constructs were expressed in either insect or mammalian cells and were identified as a mixture of heterodimeric and higher-order oligomers. Structure-based design of a soluble E1E2 heterodimer provided an opportunity to characterize this assembly with unprecedented detail. Along with evidence of heterogeneity, Cao et al. demonstrated binding of several neutralizing antibodies to both E1E2 constructs, though the antibodies were limited to those that target E2 domains B or E, or an E1 epitope. Expression of stable E1E2 constructs also allowed for low-resolution electron microscopy reconstructions of E1E2 with the designed heterodimeric scaffold, which were compared to a computational prediction of the E1E2 glycoprotein structure generated using E1E2 sequence co-evolution [124]. Another study reported the design of several constructs that replaced E1E2 transmembrane domains with scaffolds, in conjunction with a furin cleavage site to permit expression of the designs as a cleaved polyprotein, identifying a design with a human leucine zipper scaffold (sE1E2.LZ) that displayed native-like antigenicity and immunogenicity [107]. Though this construct was somewhat heterogeneous based on analytical characterization, it was shown to be smaller and less complex than full-length E1E2 that was extracted from the cell membrane with intact transmembrane domains. Antigenic characterization confirmed that sE1E2.LZ bound with high affinity not only to antibodies that target E2 antigenic domains A-E and E1E2, but also to CD81. Neutralization of homologous (H77) HCVpp by sE1E2.LZ-immunized murine sera was comparable to neutralization from sera immunized with full-length E1E2, suggesting that this scaffolded E1E2 represents a secreted and native-like form of the E1E2 heterodimer [107].

Additional efforts have designed E1E2 constructs for optimal secretion and have tested for native-like properties with antibody binding assays. One study described several designs of secreted E1E2 that removed E1 and E2 transmembrane domains and connected the ectodomains with various tags and linkers, including one design with a cleavage site between E1 and E2 [125]. These designs showed binding to multiple antibodies targeting different epitopes, but an sE1E2 construct in a DNA format induced only limited serum neutralization in mice [125]. Another effort expressed and purified E1E2 with the C-terminal oligomerization domain of C4b-binding protein, resulting in E1E2 heptamers that elicited immune responses from mice with some capacity to neutralize HCVpp [100].

### 2.4. New and Alternative Approaches

While a number of rational E2, E1E2, and epitope-based designs have been tested, recent studies have yielded new structural and mechanistic insights regarding HCV immune evasion and entry that may inform prospective designs. Notably, the impact of conformational flexibility in E2 regions, especially HVR1 and domain E, on antibody resistance and receptor binding has been explored [34,126,127]. In comparing resistant sequences with the sensitive reference sequence H77 [128], mutations in a five-residue motif in HVR1 were found to directly contribute to differences in neutralization resistance, corroborating previous findings [129]. These HVR1 variants were more dependent on the SR-BI co-receptor for entry, and showed a higher propensity to adopt β-hairpin conformations in domain E. In addition, SR-BI dependency was strongly correlated with accessibility of the CD81-binding site and other bnAb epitopes, suggesting that CD81 may induce a conformational transition from closed to open in a subsequent attachment step that is dependent on the presence of HVR1 [122,130]. Closed and open states of E2 were linked to distinct conformations of domain E through temperature-dependent neutralization assays, with a β-hairpin associated with a closed state, and an extended, HC33.1 bnAb-bound epitope conformation associated with an open state [126]. With an improved mechanistic understanding of E2 conformational dynamics, structure-based vaccine designs can better account for the effect of inherent flexibility on bnAb epitope accessibility and immune evasion.

A variety of vaccines have been designed for HCV without using E1E2, instead utilizing antigens from core [131,132,133], p7 [134] or non-structural proteins [135,136], which have been evaluated for stimulation of T cell responses [137,138], with a focus on DNA vaccines [139,140,141] or expression with viral vectors [142,143]. One T cell-based vaccine with viral vectored non-structural proteins has been tested in a phase 1/2 clinical trial for protection against HCV infection in healthy at-risk individuals; while it failed to prevent chronic HCV infection, it did induce HCV-specific T cell responses [144]. Rationally designed T cell antigens, possibly in conjunction with B cell antigens, may be a means to improve efficacy of such approaches. Vaccine designs combining the HCV core protein with E1E2 can induce both B cell and T cell responses, as shown by a VLP assembly of four different HCV genotypes [145] and a chimeric protein with epitopes from all three antigens and NS3 [146]. Although p7 has not been the focus of vaccine studies until recently, overlapping peptides of the antigen displayed on nanoparticles stimulated significantly higher T cell responses in mice, providing another avenue for vaccine development [147]. Rational design approaches have also been utilized to stimulate CD8^+^ T cell responses by accounting for HCV genetic diversity. Two studies created synthetic genotype 1 sequences through mosaic and ancestral sequence methods [114,148], and were validated for in vivo immunogenicity or T cell stimulation [149,150]. Of note, the mosaic vaccine approach was previously utilized for HIV antigen design [151], and mosaic HIV antigens were found to be immunogenic in a phase 1/2a clinical trial (NCT02315703) and protective against simian-human immunodeficiency virus (SHIV) infection in rhesus macaques [152].

## 3. Discussion

The strategies of glycoprotein optimization, design, and scaffolding have produced a variety of rational and structure-based designs of HCV vaccine candidates. Collectively, these approaches have provided useful data and insights into determinants of immunogenicity and bnAb elicitation, as well as novel platforms for epitope and glycoprotein presentation. Despite this progress, the ability to conduct rational HCV vaccine design would dramatically increase with greater structural information on E1E2 glycoproteins, as demonstrated by immunogens generated using structure-based design for influenza [153,154,155], HIV [39,156], and SARS-CoV-2 [42,69,157,158], which have known glycoprotein complex structures available for reference in design studies. High-resolution structural characterization of the E1E2 heterodimer would be immensely useful in this regard, while useful insights would also be gained through the structure of E2 bound to CD81, and any component of the complex interactions between HCV LVPs and multiple lipoprotein receptors [29]. As an alternative to structural characterization of secreted, soluble E1E2 ectodomains, characterization of the native, membrane-bound E1E2 heterodimer can be investigated using methods such as cryogenic electron microscopy with lipid nanodiscs, which has already been used for characterization of full-length HIV glycoprotein assemblies [159]. 

In future studies, designed HCV antigens can utilize approaches such as nanoparticle or multivalent display of E1E2 or key epitopes, in conjunction with T cell antigen designs for other HCV proteins, to optimize immunogenicity while effectively coordinating stimulation of broad B cell and T cell responses. With multivalent HCV antigens showing promising results in at least two studies [105,145], incorporating multivalency may be advantageous for future candidates given the high sequence variability of HCV and past success with other viruses, specifically regarding the elicitation of immune responses and protection from challenge in animal models [160]. Finally, complete and consistent evaluations of vaccine designs may be impractical without overcoming current limitations in animal models and assessments of immunogenicity, including the lack of available immunocompetent animal models for HCV challenge studies, and the use of murine and guinea pig models in immunogenicity studies, as these species are unable to reflect key features of human bnAb responses to HCV [80,161] due to differences in antibody germline genes. Several of these limitations have been addressed by recent work, including the finding that E1E2 immunization in rhesus macaques can generate bnAbs with sequence and structural features that are similar to human bnAbs [162,163], and the identification and use of hepaciviruses as HCV surrogates for immunocompetent in vivo studies [164,165]. The many recent advances in HCV immunogen design and vaccine assessment, along with lessons and concepts from vaccine design efforts for other viruses and pathogens, can enable the successful development of an optimal and effective HCV vaccine.

## Data Availability

Data discussed in this review are available in the cited publications, and structural coordinates shown in Figure 1 are from the Protein Data Bank (https://www.rcsb.org (accessed on 26 April 2021)).

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
