# Peer review of "Structure-Based and Rational Design of a Hepatitis C Virus Vaccine"

_viruses, 2021, doi:10.3390/v13050837_

Round 1

Reviewer 1 Report

Despite the effective DAA drugs treatment, a safe and effective HCV vaccine is still worth to developing. This review focuses on humoral immunity and summarizes how to induce a broad range of neutralizing antibodies based on the structure of the envelop glycoproteins, and introduces T cell-based vaccine. However, Due to the lack of high-resolution structural analysis of the E1E2 heterologous proteins and the lack of animal models for evaluating virus infectivity, many discussions still depend on the recognition of the neutralizing antibody epitope. Based on these, this review is a comprehensive review of HCV vaccine design. But there are a few contents that can be improved

  1. The description of fig.1 is simple. Could you mark the corresponding amino acid positions on the membrane protein and different antigenic regions, that will help the readers understand the structure more easily;
  2. In the review, the author might suggest which vaccine will be better between mono or multi-valent HCV vaccine;
  3. Will lipo-viral particles affect the structure of HCV virus envelop proteins and the binding with neutralizing antibodies?

Author Response

We thank the reviewers for their time and for their helpful comments regarding our manuscript. To respond, we have made several modifications to the manuscript text, along with Figure 1 and Table 1. Responses to the comments of the reviewers are below, with original reviewers’ comments in italics for reference, while manuscript changes in the text to address the reviewers’ comments are in red.

Reviewer 1

Despite the effective DAA drugs treatment, a safe and effective HCV vaccine is still worth to developing. This review focuses on humoral immunity and summarizes how to induce a broad range of neutralizing antibodies based on the structure of the envelop glycoproteins, and introduces T cell-based vaccine. However, Due to the lack of high-resolution structural analysis of the E1E2 heterologous proteins and the lack of animal models for evaluating virus infectivity, many discussions still depend on the recognition of the neutralizing antibody epitope. Based on these, this review is a comprehensive review of HCV vaccine design. But there are a few contents that can be improved

  1. The description of fig.1 is simple. Could you mark the corresponding amino acid positions on the membrane protein and different antigenic regions, that will help the readers understand the structure more easily;

We thank the reviewer for this helpful suggestion regarding Figure 1. We updated Figure 1 with labels of amino acid positions for the E1E2 transmembrane domains and ectodomains. While we agree labeling of antigenic region positions on a structure could be informative, Figure 1 was intended as a broad representation of design strategies rather than a high resolution model of the E1E2 complex structure. Since a complete E1E2 heterodimeric structure is unknown and structures of individual ectodomains are incomplete, we felt that labeling specific antigenic region locations in the context of our E1E2 schematic representation may be confusing or potentially misleading to readers, so we chose not to do that.

  1. In the review, the author might suggest which vaccine will be better between mono or multi-valent HCV vaccine;

We agree that monovalency vs. multivalency should be addressed. Accordingly sentences were added to the Discussion section (lines 332-339):

“In future studies, designed HCV antigens can utilize approaches such as nanoparticle or multivalent display of E1E2 or key epitopes, in conjunction with T cell antigen designs for other HCV proteins, to optimize immunogenicity while effectively coordinating stimulation of broad B cell and T cell responses. With multivalent HCV antigens showing promising results in at least two studies [106, 146], incorporating multivalency may be advantageous for future candidates given the high sequence variability of HCV and past success with other viruses, specifically regarding the elicitation of immune responses and protection from challenge in animal models [162].”

  1. Will lipo-viral particles affect the structure of HCV virus envelop proteins and the binding with neutralizing antibodies?

The role of lipo-viral-particles in immune evasion, particularly through modulation of neutralizing antibody responses, has been addressed in previous studies. The manuscript text was updated to note how lipo-viral-particles have contributed to the difficulties of HCV vaccine development, along with references, in the Introduction section (lines 35-40):

“Multiple factors likely contribute to the difficulty in developing an HCV vaccine [20, 21], including substantial diversity between genotypes [22, 23], viral mutation in infected individuals leading to quasispecies that can escape neutralizing antibodies [24], epitope shielding by glycans on the E1 and E2 envelope proteins [25, 26], epitope shielding by apolipoproteins in HCV lipo-viral-particles (LVPs) [27-29], and other mechanisms of immune evasion [30, 31].”

Reviewer 2 Report

This is a nice review on the state-of-the art of structure-based design of candidate HCV vaccines.  The work is clearly written, well documented, and provides a good summary of current research on envelope-based antigen vaccine candidates.  The following are a few suggestions that might strengthen the review and provide insights into some of the challenges of HCV vaccine design.  Specifically:

  1. The introduction raises many of the challenges to HCV vaccine design, but does not mention two key difficulties.
    A.  First, the composition of HCV particles in humans is poorly understood, and HCVpps and HCVccs may not be representative of these particles that appear highly interactive (or even within) VLDL and LDL particles (see P. Andre et al, JVI 2002 and others). These lipo-viro-particles likely shield E1/E2 heterodimers and/or alter presentation of antigenic regions in vivo.
    B.  Second, the assays used to measure neutralization should be described and clarified in the text, including animal models. Neutralization of HCVpp and HCVcc have major limitations, different manuscripts utilize different methods, and animal testing is not well standardized.  The author notes in one section that mice injected with one antigen were protected against challenge by genotypes 1 and 2 (lines 139-141).  It would be helpful for the authors to describe the animal model – whether or not the mouse is immune suppressed (i.e. RAG-/-), and how this may influence vaccination study results.

While these limitations are important, they do not negate the usefulness of current methods and vaccine design, but to provide readers not familiar with the limitations of HCV replication models (in vitro and in vivo), this information is needed.

  1. I agree that understanding E1/E2 structure may enhance rationale design of HCV candidate antigens, the comment that “the path to an effective HCV vaccine may include utilization of successful design strategies employed for other viruses” and here (line 72-75, 81-82, 293-296) and later note HIV, RSV, influenza as examples. It is true that good structural data have led to the design of rationale candidate vaccine antigens; however, it seems overstated to call this “successful design strategies” when there are not effective HIV or universal influenza vaccines.  These two statements should be more clearly described.
  2. In lines 272-274, it should be noted the type of studies referenced? Are these testing for protection, treatment of chronically viremic subjects, or both?  What are the endpoints for these clinical trials and how did the results achieve or fail to reach these?
  3. Table 1 provides a great deal of useful information. Another column providing the results of testing, and the testing methods would make this more interesting.
  4. Development of an HCV vaccine would be a great achievement, and this review summarizes the state of E1/E2 structure-design research well. There should be a bit more information on the limitations of current systems to evaluate vaccines, and how these challenges could be approached going forward.

Author Response

We thank the reviewers for their time and for their helpful comments regarding our manuscript. To respond, we have made several modifications to the manuscript text, along with Figure 1 and Table 1. Responses to the comments of the reviewers are below, with original reviewers’ comments in italics for reference, while manuscript changes in the text to address the reviewers’ comments are in red.

This is a nice review on the state-of-the art of structure-based design of candidate HCV vaccines.  The work is clearly written, well documented, and provides a good summary of current research on envelope-based antigen vaccine candidates.  The following are a few suggestions that might strengthen the review and provide insights into some of the challenges of HCV vaccine design.  Specifically:

  1. The introduction raises many of the challenges to HCV vaccine design, but does not mention two key difficulties.
    A.  First, the composition of HCV particles in humans is poorly understood, and HCVpps and HCVccs may not be representative of these particles that appear highly interactive (or even within) VLDL and LDL particles (see P. Andre et al, JVI 2002 and others). These lipo-viro-particles likely shield E1/E2 heterodimers and/or alter presentation of antigenic regions in vivo.

We agree that lipo-viral-particle formation and composition is an important challenge for vaccine development that should be mentioned. The Introduction section (lines 35-40) has been modified to address this point, which was also raised by Reviewer 1:

“Multiple factors likely contribute to the difficulty in developing an HCV vaccine [20, 21], including substantial diversity between genotypes [22, 23], viral mutation in infected individuals leading to quasispecies that can escape neutralizing antibodies [24], epitope shielding by glycans on the E1 and E2 envelope proteins [25, 26], epitope shielding by apolipoproteins in HCV lipo-viral-particles (LVPs) [27-29], and other mechanisms of immune evasion [30, 31].”

  1. Second, the assays used to measure neutralization should be described and clarified in the text, including animal models. Neutralization of HCVpp and HCVcc have major limitations, different manuscripts utilize different methods, and animal testing is not well standardized.  The author notes in one section that mice injected with one antigen were protected against challenge by genotypes 1 and 2 (lines 139-141).  It would be helpful for the authors to describe the animal model – whether or not the mouse is immune suppressed (i.e. RAG-/-), and how this may influence vaccination study results.

While these limitations are important, they do not negate the usefulness of current methods and vaccine design, but to provide readers not familiar with the limitations of HCV replication models (in vitro and in vivo), this information is needed.

We agree that methods of in vitro and in vivo testing have limitations and are not standardized, and are components of HCV vaccine candidate evaluation that pose additional challenges to development. This point has been noted in the Introduction (lines 40-42):

“Current limitations of and lack of standardization for in vitro and in vivo models of HCV infection may also hinder the evaluation and comparison of vaccine candidates [13, 32].”

The recent Cowton et al. study uses an anti-idiotype method unusual for HCV vaccine design, so we agree that more detail is warranted. The mouse model used for HCV challenge is immunosuppressed, and the text has been changed accordingly (lines 153-156):

“Antibodies induced by B2.1A protected against HCV challenge from genotype 1 or 2 strains in an immunosuppressed and humanized liver chimeric mouse model, suggesting that B2.1A could form the basis of distinct immunogen designs focused on antigenic domain E [84]. ”

  1. I agree that understanding E1/E2 structure may enhance rationale design of HCV candidate antigens, the comment that “the path to an effective HCV vaccine may include utilization of successful design strategies employed for other viruses” and here (line 72-75, 81-82, 293-296) and later note HIV, RSV, influenza as examples. It is true that good structural data have led to the design of rationale candidate vaccine antigens; however, it seems overstated to call this “successful design strategies” when there are not effective HIV or universal influenza vaccines.  These two statements should be more clearly described.

These statements regarding successful design strategies have been edited for clarification, and the updated text is noted below.

Lines 81-84 (formerly lines 72-75):

“The path to an effective HCV vaccine may include utilization of design strategies that generated promising immunogens for other viruses, including HIV, RSV, and SARS-CoV-2 [38-42], which in some cases led to clinical-stage candidates [65-67] and successful vaccines [68, 69].”

Lines 89-91 (formerly lines 81-82):

“These strategies attempt to focus responses to those key sites and have been employed for RSV and HIV immunogen designs to elicit more neutralizing antibodies than the un-modified proteins [38, 70].”

Lines 316-320 (formerly lines 293-296):

“Despite this progress, the ability to conduct rational HCV vaccine design would dramatically increase with greater structural information on E1E2 glycoproteins, as demonstrated by immunogens generated using structure-based design for influenza [154-156], HIV [39, 157], and SARS-CoV-2 [69, 158-160], which have known glycoprotein complex structures available for reference in design studies.

  1. In lines 272-274, it should be noted the type of studies referenced? Are these testing for protection, treatment of chronically viremic subjects, or both?  What are the endpoints for these clinical trials and how did the results achieve or fail to reach these?

We revised and expanded the description of this recent clinical trial in the text to address these questions (now lines 296-301):

“One T cell-based vaccine with viral vectored non-structural proteins has been tested in a phase 1/2 clinical trial for protection against HCV infection in healthy at-risk individuals; while it failed to prevent chronic HCV infection, it did induce HCV-specific T cell responses [145]. Rationally designed T cell antigens, possibly in conjunction with B cell antigens, may be a means to improve efficacy of such approaches.”

  1. Table 1 provides a great deal of useful information. Another column providing the results of testing, and the testing methods would make this more interesting.

To add more information about the studies cited to Table 1, we included a “Results” column that briefly summarizes the results of testing for neutralization. Testing methods were either HCVpp or HCVcc neutralization assays and labeled as such for each study.

  1. Development of an HCV vaccine would be a great achievement, and this review summarizes the state of E1/E2 structure-design research well. There should be a bit more information on the limitations of current systems to evaluate vaccines, and how these challenges could be approached going forward.

We agree that the limitations of current methods for vaccine evaluation should be addressed. We added more information regarding some of these challenges and how they are being approached in the Discussion (lines 339-348):

“Finally, complete and consistent evaluations of vaccine designs may be impractical without overcoming current limitations in animal models and immunogenicity assessment, including the lack of available immunocompetent animal models for HCV challenge studies, and the use of murine and guinea pig models in immunogenicity studies, as these species are unable to reflect key features of human bnAb responses to HCV [81, 163] due to differences in antibody germline genes. Several of these limitations have been addressed by recent work, including the finding that E1E2 immunization in rhesus macaques can generate bnAbs with sequence and structural features that are similar to human bnAbs [164, 165], and the identification and use of hepaciviruses as HCV surrogates for immunocompetent in vivo studies [166, 167].”

Reviewer 3 Report

Guest and Pierce presented an interesting review about Structure-based and rational design of a hepatitis C virus vaccine. Teh review includes the following items: epitope scaffolding and epitope-based designs,  E1E2 optimization, antigen design, and other  approaches. Also, the authors provided a table showing the use of  E2 and E1E2 immunogen designs tested in vivo animal models.

In general the review is very interesting, has an acceptable flow and should be published.

Author Response

We thank the reviewers for their time and for their helpful comments regarding our manuscript. To respond, we have made several modifications to the manuscript text, along with Figure 1 and Table 1. Responses to the comments of the reviewers are below, with original reviewers’ comments in italics for reference, while manuscript changes in the text to address the reviewers’ comments are in red.

Guest and Pierce presented an interesting review about Structure-based and rational design of a hepatitis C virus vaccine. Teh review includes the following items: epitope scaffolding and epitope-based designs, E1E2 optimization, antigen design, and other approaches. Also, the authors provided a table showing the use of E2 and E1E2 immunogen designs tested in vivo animal models.

In general the review is very interesting, has an acceptable flow and should be published.

We thank the reviewer for these positive comments.